# The Mechanism of Bisdemethoxycurcumin Enhances Conventional Antibiotics against Methicillin-Resistant *Staphylococcus aureus*

**DOI:** 10.3390/ijms21217945

**Published:** 2020-10-26

**Authors:** Shu Wang, Min-Chul Kim, Ok-Hwa Kang, Dong-Yeul Kwon

**Affiliations:** Department of Oriental Pharmacy, College of Pharmacy and Wonkwang Oriental Medicines Research Institute, Wonkwang University, Iksan 54538, Jeonbuk, Korea; wshu1996@gmail.com (S.W.); omp1004@naver.com (M.-C.K.); sssimi@wku.ac.kr (D.-Y.K.)

**Keywords:** MRSA, bisdemethoxycurcumin, β-lactam antibiotics, synergy, *mecA*, PBP2a

## Abstract

Methicillin-resistant *Staphylococcus aureus* (MRSA) infection has posed a serious threat to public health, therefore, the development of new antibacterial drugs is imperative. Bisdemethoxycurcumin (BDMC) is a curcumin analog that exists in nature and possesses extensive pharmacological actions. This review focuses on investigating the antibacterial activity of BDMC alone or in combination with three antibiotics against MRSA. We determined the minimal inhibitory concentration of BDMC, with a broth microdilution assay, and the value against all six strains was 7.8 μg/mL. The synergistic effect of BDMC combined with the antibiotics was determined using a checkerboard dilution test and a time–kill curve assay. The results showed that the antimicrobial effect of BDMC combined with antibiotics was superior to treatment with that of a single agent alone. We examined the antibacterial activity of BDMC in the presence of a membrane-permeabilizing agent and an ATPase-inhibiting agent, respectively. In addition, we analyzed the *mecA* transcription gene and the penicillin-binding protein 2a (PBP2a) level of MRSA treated with BDMC by quantitative RT-PCR or Western blot assay. The gene transcription and the protein level were significantly inhibited. This study demonstrated that BDMC has potent antibacterial activity, and proved that BDMC may be a potential natural modulator of antibiotics.

## 1. Introduction

Methicillin-resistant *Staphylococcus aureus* (MRSA) is considered to be a global health threat and economic burden each year [1]. It is an important cause of nosocomial infections [2], and causes severe infections at diverse body sites [3,4]. The therapeutic options for patients infected with MRSA are limited due to the resistance to most ordinarily used antibiotics [5]. MRSA resistance in *Staphylococcus aureus* (*S. aureus*) and all β-lactam antibiotics are attributed to the expression of the *mecA* gene that encodes penicillin-binding protein 2a (PBP2a) [6]. Vancomycin has emerged as the latest antibiotic to treat MRSA. However, overprescription of vancomycin has disrupted the susceptibility of *S. aureus* strains to vancomycin [7]. Antibiotic resistance becomes the main burden in the treatment of bacterial infections; hence, new antimicrobial agents and therapies are urgently needed [8].

In recent years, natural antimicrobials have become a promising approach and an alternative to antibiotics because of their efficacy and tendency to develop bacterial resistance [9]. Bisdemethoxycurcumin (BDMC) is a component of turmeric, the bright yellow spice, derived from the roots of *Curcuma longa* L. (*C. longa* L.) [10]. *C. longa* L. is a major component of the Indian spice turmeric and is used frequently as a dietary supplement or herbal medicine [11]. *C. longa* L. mainly consists of three analogs, i.e., curcumin, demethoxycurcumin, and BDMC, which exhibit anti-inflammatory, antioxidative, and antibacterial activities [12]. BDMC has a variety of pharmacological activities, including antitumor [13], protecting cardiomyocyte [14], and inhibiting allergic diseases [15]. However, its inhibiting role in MRSA has not been reported. In the present study, we investigated the antibacterial activity of BDMC and the mechanism of synergistic activity among BDMC and three conventional antibiotics against MRSA at the molecular level.

## 2. Results

### 2.1. Antimicrobial Susceptibility Testing and Synergic Effect

The minimal inhibitory concentration (MIC) value of BDMC against all six *S. aureus* strains was 7.8 μg/mL (Table 1). The antimicrobial susceptibility of MRSA strains to BDMC is presented in Table 2. It shows that the growth of MRSA was strongly inhibited. In the present study, checkerboard assays were performed to evaluate the synergy effect of BDMC in combination with the three antibiotics. The MIC vales of gentamicin, ampicillin, and oxacillin were reduced from two-fold to 16-fold, two-fold to eight-fold, and two-fold to four-fold, respectively (Table 2). Among them, the synergy effect of BDMC with gentamicin was the most prominent, and the fractional inhibitory concentration index (FICI) was as low as 0.1.

### 2.2. Time–Kill Curve Assay

The synergy effects of BDMC in combination with the three antibiotics against the *S. aureus* strains were further evaluated by the time–kill curve assay. The growth curves of the DPS-1 strain show that the supplementation of 1/2 MIC BDMC with 1/2 MIC gentamicin resulted in a marked reduction in the growth of MRSA after 4 h incubation time, with complete growth inhibition after 8 h incubation time (Figure 1a). When 1/2 MIC BDMC was supplemented with 1/2 MIC ampicillin, a marked reduction was observed in the growth of ATCC 33591 strain following 8 h incubation time and induced rapid inhibition in a time-dependent manner (Figure 1b). The growth curves of CCARM 3090 strain show that the supplementation of 1/2 MIC BDMC with 1/2 MIC oxacillin resulted in a marked reduction in the growth of MRSA following 8 h incubation time, and induced rapid inhibition in a time-dependent manner within 24 h incubation time (Figure 1c). The combination of BDMC with antibiotics caused more than 3 log_10_ CFU/mL reductions on all the three *S. aureus* strains as compared with the three antibiotics alone.

### 2.3. Antimicrobial Activity with the ATPase Inhibitor and the Membrane-Permeabilizing Agent

The antimicrobial activity of BDMC with the ATPase inhibitor (DCCD, N,N′-dicyclohexylcarbodiimide) and the membrane-permeabilizing agent (TX-100, Triton X-100) was tested. The influence of the membrane-permeabilizing agent and the ATPase inhibitor on the susceptibility of MRSA to BDMC is presented in Figure 2. DCCD reduces the ATP levels by disrupting the electrochemical proton gradient, and TX-100 has the capacity to increase the permeability of the outer membrane. In comparison with the optical density at 600 nm (OD 600) value of the control, the OD 600 value of the suspension was significantly decreased, and the bacterial viability under the combination of 1/16 MIC BDMC (0.45 µg/mL) and 125 µg/mL DCCD was reduced by 43.5% (Figure 2a). In comparison with the OD 600 value of 1/32 MIC BDMC alone (0.225 µg/mL), the OD 600 value of the suspension in the presence of 1/32 MIC BDMC and 0.00001% TX-100 was reduced by 0.33% (Figure 2b).

### 2.4. Expression of PBP2a in Methicillin-Resistant Staphylococcus aureus (MRSA)

Western blotting was performed to detect the protein expression of PBP2a in MRSA. PBP2a expression levels under the tested treatments are summarized in Figure 3. The experimental samples were composed of control (Lane 1), 1/8 MIC BDMC (Lane 2), 1/4MIC (Lane 3), and 1/2 MIC BDMC (Lane 4). The control was not treated with BDMC. As shown in the figure, the protein level was reduced as BDMC was added at the concentrations specified. Especially with 1/2 MIC BDMC (3.9 µg/mL), the PBP2a level was reduced markedly. The data indicated that the addition of BDMC reduced the PBP2a level in a dose-dependent manner.

### 2.5. BDMC Represses the Transcription of MecA, BlaZ, BlaR1, and MecR1 in S. aureus

The transcriptional levels of *blaZ*, *blaR1*, *mecA,* and *mecR1* were inhibited in *S. aureus* upon the treatment with 1/8 MIC (0.9 μg/mL) concentrations of BDMC, and the transcription of four genes was affected by the treatment with graded subinhibitory concentrations (Figure 4a–d). In the presence of 1/2 MIC (3.9 μg/mL) of BDMC, the transcriptional levels of *blaZ*, *blaR1*, *mecA*, and *mecR1* were reduced by 1.9-fold, 2.6-fold, 2.7-fold, and 2.6 fold, respectively.

## 3. Discussion

The antimicrobial susceptibility of MRSA strains to BDMC, either alone or in combination with antibiotics, showed potent activity. The MIC of BDMC was as low as 7.8 μg/mL, and its antibacterial effect was significantly better than that of the three reference conventional antibiotics. The MICs were 16- to 32-folds lower than the results (125 to 250 μg/mL) of our previous study on curcumin against MRSA [16]. In addition, BDMC was found to be the most stable of the three turmeric analogs [17], revealing BDMC as an exceptionally promising antibacterial substance. However, potential antimicrobial substances would be with potent antibacterial activity and would also restore the antibacterial sensitivity of existing antibiotics [3]. Synergistic activity was investigated in combination with subinhibitory concentrations of conventional antibiotics to study the reversal ability of BDMC to antibiotics. The data showed that BDMC in combination with the three antibiotics separately all had a synergy or partial synergy effect against six *S. aureus* strains, and significantly improved the sensitivity of gentamicin. Thus, this study, firstly, demonstrated the synergistic activity of BDMC and antibiotics against MRSA infections in vitro.

The time–kill assay was used to further determine the synergistic antimicrobial effect. Contrasted with the control group, the sub-concentration group showed almost unchanged log10 CFU/mL of BDMC after 24 h, indicating that BDMC could inhibit the growth of MRSA. The result showed that BDMC had synergy effects in combination with certain antibiotics. The log10 CFU/mL of the three strains all decreased significantly at 8 h. It might be attributed to the polyphenol structure of BDMC. The polyphenol structure can destroy the cell wall of bacteria [18], thus, increasing the efficiency of antibiotics entering the cell, and then enhancing the antibacterial effect.

We examined the antibacterial activity of BDMC in the presence of membrane function. DCCD is known to decrease the ATP levels by disrupting the electrochemical proton gradient [19], and TX-100 has the capacity to increase the permeability of the outer membrane [20]. The data showed that the antibacterial activity of BDMC was significantly enhanced in the presence of DCCD or TX-100. The results demonstrate that the anti-MRSA activity of BDMC was enhanced by changes in membrane permeability and the decrease in the ATP level.

Partial synergy effects were also detected when BDMC combined with the β-lactam antibiotics oxacillin was applied to the treatment against MRSA. The authors demonstrated the mechanism of action of BDMC combined with oxacillin in the treatment against MRSA by studying the *mecA* gene and PBP2a. As mentioned earlier, resistance to β-lactam antibiotics is due to MRSA carrying the *mecA* gene that encodes PBP2a with low β-lactam affinity [21]; in addition, *blaZ, blaR1,* and *mecR1* participate in the regulation of *mecA* [22]. The altered levels of PBP2a and the transcriptional regulation of *mecA* declined significantly after the treatment with BDMC. It was speculated that BDMC interfered *blaR1* to promote the signal transduction pathway of *mecA* activation, and then affected the synthesis of PBP2a. Western blotting was performed to detect the protein expression of PBP2a in MRSA. The results are consistent with previous assumptions, the protein level was reduced when BDMC was added at the concentrations specified. Especially with 1/2 MIC BDMC (3.9 µg/mL), the PBP2a level was reduced markedly. The results indicated that BDMC inhibited the transcriptional and translational expression levels of *mecA*.

The data of quantitative RT-PCR (qRT-PCR) indicated that BDMC repressed the transcription of *mecA, blaZ, blaR1,* and *mecR1* in *S. aureus.* The expression of all four genes was significantly reduced. The above results are consistent with our previous assumptions that BDMC could potentiate the influence of oxacillin on MRSA. This finding is helpful to explore the potential value of combining a clinical therapeutic with a natural antibacterial therapy. In addition, we believe that it is equally important to strengthen the supervision of the use of antibiotics and publicize the consequences of the abuse of antibiotics.

Natural compounds are considered to be safe and eco-friendly [23], and have a long history as the choice of drug discovery [24]. Hence, the natural antibacterial compound becomes an attractive alternative. In the present study, we were the first to investigate the potential of the natural agent BDMC from *C. longa L.* against MRSA, as well as the synergistic activity of BDMC combined with antibiotics. This study gives evidence for the BDMC as a modulator of antibiotics and provides a reference for the study of antibiotic resistance. Although BDMC appears to be a promising natural antibacterial agent, the study has limitation. The results of the present study only suggest that BDMC has antibacterial activity in vitro. Therefore, further studies should be conducted to verify the mechanism of BDMC against MRSA in vivo, and the types of MRSA strains used in the experiment should be diversified. Moreover, additional toxicity and safety experiments are necessary for effective treatment. Additional studies will be conducted in the future to overcome these limitations.

## 4. Materials and Methods

### 4.1. Reagents

BDMC was obtained from Tokyo Chemical Industry Co., Ltd. (Tokyo, Japan). Primary antibodies to PBP2a were purchased from DiNonA Inc. (Seoul, Korea). Anti-mouse IgG secondary antibody was purchased from Thermo Scientific Inc. (Waltham, MA, USA). The chemiluminescent ECL assay kit was purchased from ATTO Corp. (Tokyo, Japan). E.Z.N.A. The bacterial RNA Kit was purchased from Omega Bio-Tek (Norcross, GA, USA). The sequences of primers used in this study were purchased from Bioneer (Daejeon, Korea) (Table 3). Mueller–Hinton agar, Mueller–Hinton broth, and skim milk were obtained from Difco Laboratories (Baltimore, MD, USA). Ampicillin, gentamicin, oxacillin, TX-100, and DCCD were obtained from Sigma-Aldrich Co. (St. Louis, MO, USA). SMART™ bacterial protein extraction solution was purchased from Intron Bio Technology, Inc. (Seongnam, Korea). The QuantiTect Reverse Transcription Kit was purchased from (Dusseldorf, Germany). The Power SYBR Green PCR Master Mix was purchased from Life Technologies LTD (Warrington, UK).

### 4.2. Bacterial Strains and Growth Medium

A total of six MRSA strains were used in the study. ATCC 33591 was obtained from the American Type Culture Collection (Manassas, VA, USA). CCARM 3090, CCARM 3090, CCARM 3095, and CCARM 3102 strains were provided by the Culture Collection of Antimicrobial Resistant Microbes (National Research Resource Bank, Seoul, Korea). The clinical MRSA isolates DPS-1 were obtained from patients at the Wonkwang University Hospital (Jeonbuk, Korea). Mueller–Hinton agar (MHA) and Mueller–Hinton broth (MHB) were used as the culture medium for bacterial at 37 °C.

### 4.3. Susceptibility Testing of BDMC with Antibiotics

The minimal inhibitory concentration (MIC) of BDMC, either alone or in combination with the antibiotics (gentamicin, ampicillin, and oxacillin), was determined with the microdilution and checkerboard assays. Serial dilutions of BDMC with antibiotics were mixed in cation-supplemented MHB. The final bacterial concentration after inoculation was 1.5 × 10^5^ CFU/spot. In vitro interaction between the drugs was quantified by determining the fractional inhibitory concentration (FIC). The FIC index (FICI) was calculated as follows: ∑FIC: FICA + FICB = MICA + B/MICA alone + MICB + A/MICB alone. The FICI was interpreted as follows: ≤0.5 synergy, >0.5–0.75 partial synergy, >0.75–1 additive effect, >1–4 no effect, and >4 antagonism [25]. Three independent experiments were performed, with the data presented as the mean ± standard deviation.

### 4.4. Time–Kill Curve Assay

The synergistic antimicrobial effect was further determined by a time–kill assay, according to a previous method [26]. The influences of subinhibitory concentrations of BDMC and antibiotics (gentamicin, ampicillin, and oxacillin), either as individual drugs or in combination, on the growth of ATCC 33591 were examined. Bacterial growth curves were observed at five different time phases (0, 4, 8, 16, and 24 h). Moreover, 1/2 MIC BDMC combined with 1/2 MIC antibiotic, 1/2 MIC BDMC alone, and 1/2 MIC antibiotic alone were compared with a control (drug-free) to determine the synergistic effect. Bacterial cultures were diluted with sterilized MHB to ~1.5 × 10^5^ CFU/mL and the MHB-containing bacteria were incubated at 37 °C, for 24 h. After incubation for 24 h, the number of viable cells was counted on a drug-free MHA plate to determine the rate and extent of bacterial death. Three independent experiments were performed with the data presented as the mean ± standard deviation.

### 4.5. Antibacterial Activity of BDMC in the Presence of ATPase Inhibitor or the Membrane-Permeabilizing Agent

The antibacterial activity of BDMC against MRSA was determined in the presence of the ATPase inhibitor DCCD to determine whether BDMC antibacterial activity was associated with membrane function used by ATCC 33591. The antibacterial activity of 0.45 µg/mL BDMC (1/16 MIC) in the presence of 125 μg DCCD or 0.225 µg/mL BDMC (1/32MIC) in the presence of 0.00001% TX-100 was measured via spectrophotometry (optical density at 600 nm, OD600) after incubation for 24 h. The data were presented as the mean ± standard deviation of the three independent experiments.

### 4.6. Western Blot Analysis

The MRSA (ATCC 33591) culture was grown to an OD 600 value of 0.9 in MHB and treated with various concentrations of BDMC in the Western blot analysis. Cell protein extracts were harvested after 30 min and suspended in the bacterial protein extract (iNtRON Biotechnology) containing Tris-HCI (pH 7.5). The procedures were performed according to the manufacturer’s description. Protein concentrations were measured by a Bio-Rad protein assay reagent (Bio-Rad Laboratories, Hercules, CA, USA), according to the manufacturer’s instructions. The supernatant was transferred to a new tube, and aliquots of equal protein were analyzed via sodium dodecyl sulfate-polyacrylamide gel electrophoresis (SDS-PAGE). The electrophoresed gels were transferred to Amersham HybondTM-P membranes (GE Healthcare, Piscataway, NJ, USA) for the Western blot analysis. The membranes were blocked by 5% skim milk in Tris-buffered saline with Tween-20 buffer (150 mM NaCl, 20 mM Tris-HCl, and 0.05% Tween-20, pH 7.4). After blocking, the membranes were probed with monoclonal mouse anti-PBP2a primary antibody (diluted 1:1000, DiNonA, Seoul, Korea) and re-probed with anti-mouse IgG secondary antibody (diluted 1:2000, Enzo Life Sciences, Ann Arbor, MI, USA). Then, the membranes were then treated with ECL™ Prime Western Blotting Detection reagent (GE Healthcare Life Sciences, Incheon, Korea), and the bands were visualized with an ImageQuant LAS-4000 mini chemical luminescent imager (GE Healthcare Life Sciences) [27].

### 4.7. Reverse Transcription and qRT-PCR 

Strain ATCC 33591 was grown to an OD 600 value of 0.9 in MHB and subject to subinhibitory concentrations (1/8 MIC, 1/4 MIC, and 1/2 MIC) of BDMC for 0.5 h. A control without BDMC was included. Total RNA was prepared with the Easy-RED BYF total RNA extraction kit, according to the manufacturer’s procedure (iNtRON Biotechnology, Seongnam, Korea). The RNA concentration was determined by measuring A_260_ on a NanoDrop spectrophotometer (BioTek, Winooski, VT, USA). RNA was reverse transcribed into cDNA with a cDNA synthesis kit (iNtRON Biotechnology) for first-strand cDNA synthesis, in accordance with the manufacturer’s instructions, in order to synthesize the RNA template for qRT-PCR. The primer pairs used for the qRT-PCR are presented in Table 1. The PCR was set up as follows: 10 μL of 2 × SYBR premix (Life technologies, Carlsbad, California, USA), 2 μL sample cDNA and 1 μL of each primer (10 μM), and deionized water to a total volume of 20 μL. The PCR was run with the StepOnePlus real-time PCR system (Applied Biosystems, courtaboeuf, France) [28].

### 4.8. Statistical Analysis 

Analyses were performed in triplicate and data were presented as the mean ± standard deviation. The results were statistically analyzed by an independent Scheffe’ test (SPSS software version 22.0, IBM SPSS, Armonk, NY, USA). A p-value of less than 0.05 was considered to be statistically significant.

## Figures and Tables

**Figure 1 ijms-21-07945-f001:**
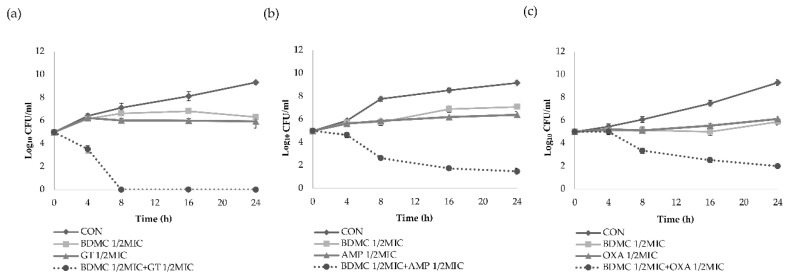
Bacterial viability at sub-minimum inhibitory concentrations of BDMC and antibiotics. (**a**) Time–kill curve of subinhibitory concentrations of 1/2 MIC gentamicin alone, 1/2 MIC BDMC alone, and 1/2 MIC gentamicin combined with 1/2 MIC BDMC against MRSA (DPS-1); (**b**) Time–kill curve of subinhibitory concentrations of 1/2 MIC ampicillin alone, 1/2 MIC BDMC alone, and 1/2 MIC ampicillin combined with 1/2 MIC BDMC against MRSA (ATCC 33591); (**c**) Time–kill curve of subinhibitory concentration of 1/2 MIC oxacillin alone, 1/2 MIC BDMC alone, and 1/2 MIC oxacillin combined with 1/2 MIC BDMC against MRSA (CCARM 3090). The data were presented as the mean ± standard deviation of the three independent experiments. CON, untreated control MRSA; GT, gentamicin; AMP, ampicillin; OXA, oxacillin; CFU, colony-forming units; MIC, minimum inhibitory concentration.

**Figure 2 ijms-21-07945-f002:**
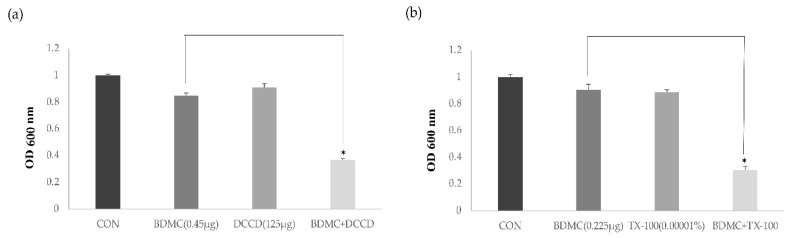
The effect of membrane-permeabilizing agents and ATPase inhibitors on the susceptibility of MRSA (ATCC 33591) to BDMC. Bacterial viability was determined by absorbance at 600 nm after incubation for 24 h with BDMC concentrations of 0.45 µg/mL BDMC in the presence of 125 μg DCCD (**a**) and 0.225 µg/mL BDMC in the presence of 0.00001% TX-100 (**b**). The data were presented as the mean ± standard deviation of the three independent experiments. * represents *p* < 0.05. CON, untreated control MRSA; DCCD, N,N′-dicyclohexylcarbodiimide; TX-100, Triton X-100; OD 600, optical density at 600 nm.

**Figure 3 ijms-21-07945-f003:**
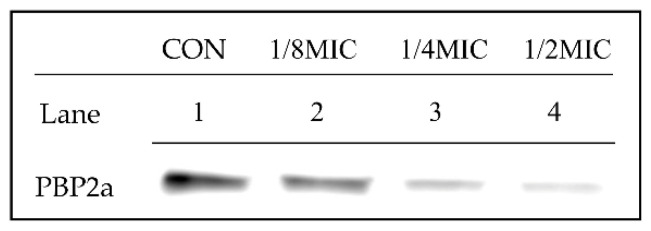
The expression of PBP2a in MRSA (ATCC 33591) cultures in the presence of various concentrations of BDMC. The PBP2a production was reduced significantly after exposure to *S. aureus* strains with 1/8 MIC BDMC (Lane 2), 1/4 MIC BDMC (Lane 3) and 1/2 MIC BDMC (Lane 4). CON, untreated control MRSA. (Lane 1).

**Figure 4 ijms-21-07945-f004:**
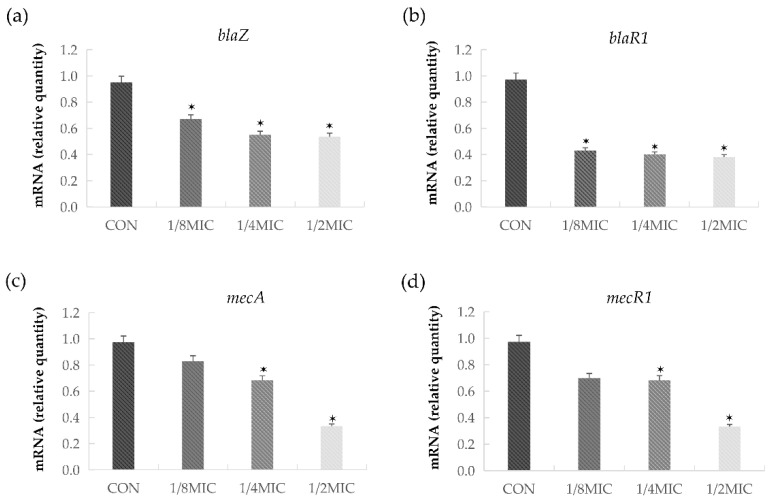
The relative gene expressions of *blaR1, blaZ, mecA,* and *mecR1* in *S. aureus* after growth at sub-concentrations of BDMC. The relative gene expressions of (**a**) *blaR1;* (**b**) *blaZ;* (**c**) *mecA;* (**d**) *mecR1* were reduced in a dose-dependent manner. The data were presented as the mean ± standard deviation of the three independent experiments. * represents *p* < 0.05. CON, untreated control MRSA.

**Table 1 ijms-21-07945-t001:** The minimal inhibitory concentration (MIC) of bisdemethoxycurcumin (BDMC) and antibiotics against methicillin-resistant *Staphylococcus aureus* (MRSA).

Strains	MIC (μg/mL)			
	BDMC	GT	AMP	OXA
ATCC 33591	7.8	3.9	62.5	250
CCARM 3090	7.8	62.5	31.3	125
CCARM 3091	7.8	250	62.5	1000
CCARM 3095	15.6	125	31.3	250
CCARM 3102	7.8	125	15.6	250
DPS-1	7.8	125	62.5	250

MIC; minimal inhibitory concentration; BDMC, bisdemethoxycurcumin; MRSA, methicillin-resistant *Staphylococcus aureus*; GT, gentamicin; AMP, ampicillin; OXA, oxacillin.

**Table 2 ijms-21-07945-t002:** The synergy effect of BDMC in combination with antibiotics.

*S. aureus* Strains	Combination with GT			Combination with AMP			Combination with OXA		
	Fold	FICI	Interpretation	Fold	FICI	Interpretation	Fold	FICI	Interpretation
ATCC 33591	4	0.75	partial synergy	4	0.75	partial synergy	2	0.51	partial synergy
CCARM 3090	8	0.2	synergy	4	0.75	partial synergy	4	0.75	partial synergy
CCARM 3091	8	0.3	synergy	2	1	additive effect	4	0.75	partial synergy
CCARM 3095	8	0.36	synergy	4	0.75	partial synergy	4	0.62	partial synergy
CCARM 3102	4	0.5	synergy	8	0.62	partial synergy	4	0.62	partial synergy
DPS-1	16	0.1	synergy	2	0.51	partial synergy	2	1	additive effect

BDMC, bisdemethoxycurcumin; GT, gentamicin; AMP, ampicillin; OXA, oxacillin; MIC, minimal inhibitory concentration; Fold, the fold of antibiotic MIC reduction; FICI, fractional inhibitory concentration index. Index interpretation: ≤0.5 synergy, 0.5–0.75 partial synergy, 0.75–1 additive effect, 1–4 no effect, and >4 antagonism.

**Table 3 ijms-21-07945-t003:** Primers used in quantitative RT-PCR (qRT-PCR).

Primer	Sequence (5′-3′)
16S RNA	F:ACTCCTACGGGAGGCAGCAG
	R:ATTACCGCGGCTGCTGG
mecA	F:CAATGCCAAAATCTCAGGTAAAGTG
	R:AACCATCGTTACGGATTGCTTC
mecR1	F:GTGCTCGTCTCCACGTTAATTCCA
	R:GACTAACCGAAGAAGTCGTGTCAG
blaR1	F:CACTATTCTCAGAATGACTTGGT
	R:TGCATAATTCTCTTACTGTCATG
blaZ	F:GCTTTAAAAGAACTTATTGAGGCTTC
	R:CCACCGATYTCKTTTATAATTT

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
