# Peer review of "The Mechanism of Bisdemethoxycurcumin Enhances Conventional Antibiotics against Methicillin-Resistant Staphylococcus aureus"

_ijms, 2020, doi:10.3390/ijms21217945_

Round 1

Reviewer 1 Report

The review of ijms-959073, In vitro antibacterial activity of bisdemethoxycurcumin and resensitizes methicillin-resistant Staphylococcus aureus to antibiotics

The title sound not very good in English. In the second sentences the subject is missing. The title should be corrected. The whole manuscript should be checked by a native English speaker.

R27: define MRSA, even if is a well-known abbreviation

R28: check the reference

R38-40: avoid the confusion on turmeric. The authors presented it as the spice, as well as the plant.

R43: “inhibitory effects of allergic diseases” sounds strange. Correct it! The whole manuscript should be checked by a native English speaker.

R44: inhibition of MRSA is very vague. It means growth inhibition?

R46: “mechanism of BDMC resensitizes methicillin-resistant Staphylococcus aureus to three conventional antibiotics, includes two β-lactam antibiotics”. The abbreviation MRSA should be used. Very bad grammar! (there are many other examples, I’m not going to list them)

R79: explain what “CON” means

R106: CON is not used in figure 3, remove the explanation

R117: present the bacterial strain used. The same for Figure 5 and for the section 107-113 and 120-124.

R205: use the journal’s style of mathematical formulas

R225: declare the source of each chemical agent and any other materials

Author Response

Answer to the comments of reviewer #1

The review of ijms-959073, In vitro antibacterial activity of bisdemethoxycurcumin and resensitizes methicillin-resistant Staphylococcus aureus to antibiotics
The title sound not very good in English. In the second sentences the subject is missing. The title should be corrected. The whole manuscript should be checked by a native English speaker.
Response:The title has been corrected. (Line:2-4)
The whole manuscript has been checked by a native English speaker.

R27: define MRSA, even if is a well-known abbreviation
Response: MRSA has been defined:
Abstract: Methicillin-resistant Staphylococcus aureus (MRSA)... (Line:10)
Introduction: Methicillin-resistant Staphylococcus aureus (MRSA)... (Line:28)

R28: check the reference
Response:Reference has been replaced. (Line:798)

R38-40: avoid the confusion on turmeric. The authors presented it as the spice, as well as the plant.
Response:Turmeric has been changed to C. longa L..(Line:40)

R43: “inhibitory effects of allergic diseases” sounds strange. Correct it! The whole manuscript should be checked by a native English speaker.
Response: The sentence has been written to ‘and inhibiting allergic diseases’ (Line:67)

R44: inhibition of MRSA is very vague. It means growth inhibition?
Response: The authors think that the growth was inhibited, and this conclusion was drawn from the comparison of the sub-inhibitory concentrations with the control group in MIC test and time killing test. (Line:283-285)

R46: “mechanism of BDMC resensitizes methicillin-resistant Staphylococcus aureus to three conventional antibiotics, includes two β-lactam antibiotics”. The abbreviation MRSA should be used. Very bad grammar! (there are many other examples, I’m not going to list them)
Response: Grammar and abbreviations have been corrected. (Line:69)

R79: explain what “CON” means
Response: “CON” have been explained (Figure.  (Line:133)

R106: CON is not used in figure 3, remove the explanation
Response: ? CON is used in figure 3 ( now is figure 2). (Line:208)

R117: present the bacterial strain used. The same for Figure 5 and for the section 107-113 and 120-124.
The bacterial strain used(ATCC 33591) has been added to the legend. (Line:236)
The section 120-124 has been rewritten. (Line:244-245)

R205: use the journal’s style of mathematical formulas
Response: (Line:700)

R225: declare the source of each chemical agent and any other materials
Response:  Sources of other chemicals agent and any other materials have been added. (Line:666-671)

We have corrected all the points pointed out by reviewer1. Please refer to the attached ijms-959073-Track Changes file.

Supplementary

1. Two errors were corrected according to the original data in Table 3.(The result of ATCC 33591 and DPS-1) (Line:83)

2. The English of the whole manuscript have been modified.

3. Through the Open Review, the authors learned that the paper must be improved in the following aspects: introduction, methods, results, and conclusions.

In addition to those listed above, the author has made the following modifications to the article:
Line: 245-279 The synergy is described more clearly.
Line: 283-289, 305-308 The results of time-kill assay and Western blot were further discussed.

Acknowledgments                                            
We would like to extend our sincere gratitude to the editorial office and two reviewers, for your instructive advice and useful suggestions on our thesis.

Reviewer 2 Report

I marked in yellow my point of view.

Author Response

Answer to the comments of reviewer #2

All of the points pointed out by reviewer2 in the pdf have been corrected. Please refer to the attached ijms-959073-Track Changes file.

Round 2

Reviewer 1 Report

The authors responded to the suggestions and improved their manuscript. Still, the authors need to check theconsistency of their abbreviations and the overall style of presenting data.

Author Response

To your judges.
I would like to express my deep gratitude to the committee for revising the paper with great consideration.
I modified it on each line. I look forward to your positive judgment.

Author's Notes to Reviewer 1:
The presentation of the data was adjusted according to the overall style:
Line 595: Figure 2 has been adjusted for the layout of (a) and (b).
In all Figures, A, B, C, and D are changed to (a), (b), (c), and (d).
Headings have been written using title case, that the first letter of all words is capitalized with the exception of short words.
The Table was renamed in the order in which it appeared. (Line 260, Line 263, Line 1163)
The resolution of all Figures has been changed to 600 dpi.
Table and figure remove bold throughout the manuscript.
The presentation of references was corrected: journal titles were changed to italics and the year was bold.

Make abbreviations consistent:
Line 42: Bisdemethoxycurcumin changed to BDMC.
Line 249: The abbreviation MIC has been added.

Supplement:
The spelling has been corrected:
Line:33 encodes; Line: 234 effect; Line:1233 qRT-PCR

Line 6-9: The punctuation has been corrected in the author's information.
Line 22: quantitative RT-PCR: Deleted the definition of unnecessary abbreviations in the abstract to keep it within 200 words.
Line 1120-1122: A new project has been added to Acknowledgments.
Line 39: Deleted (Curcuma longa), an unnecessary word.
Line: 821 Other fonts were removed.

Acknowledgments
We would like to extend our sincere gratitude to the editorial office and reviewer, for your instructive advice and useful suggestions on our thesis.

Reviewer 2 Report

Dear Colleague!

I accept in present form.

Best regards.

Author Response

(The authors gave the same response as above.)
